# Characteristics and Treatment Outcomes among Migrants with Pulmonary Tuberculosis: A Retrospective Cohort Study in Japan, 2009–2018

**DOI:** 10.3390/ijerph191912598

**Published:** 2022-10-02

**Authors:** Sangnim Lee, Myo Nyein Aung, Lisa Kawatsu, Kazuhiro Uchimura, Reiko Miyahara, Jin Takasaki, Akihiro Ohkado, Motoyuki Yuasa

**Affiliations:** 1Department of Global Health Research, Graduate School of Medicine, Juntendo University, Bunkyo-ku, Tokyo 113-8421, Japan; 2Department of Epidemiology and Clinical Research, The Research Institute of Tuberculosis, Japan Anti-Tuberculosis Association, Kiyose City, Tokyo 204-8533, Japan; 3Disease Control and Prevention Center, National Center for Global Health and Medicine, Shinjuku-ku, Tokyo 162-8655, Japan; 4Global Health Service, Faculty of International Liberal Arts, Juntendo University, Bunkyo-ku, Tokyo 113-8421, Japan; 5Advanced Research Institute for Health Sciences, Juntendo University, Bunkyo-ku, Tokyo 113-8421, Japan; 6Center for Surveillance, Immunization, and Epidemiologic Research, National Institute of Infectious Diseases, Shinjuku-ku, Tokyo 162-8640, Japan; 7Genome Medical Science Project, The Research Institute, National Center for Global Health and Medicine, Shinjuku-ku, Tokyo 162-8655, Japan; 8Department of Respiratory Medicine, National Center for Global Health and Medicine, Shinjuku-ku, Tokyo 162-8655, Japan

**Keywords:** migrant, overseas-born, foreign-born, tuberculosis, treatment outcome, Japan

## Abstract

This study aimed to describe characteristics and treatment outcomes of overseas-born pulmonary tuberculosis (PTB) patients in Japan, and identify the factors associated with “treatment non-success”. We conducted a retrospective analysis of overseas-born patients with drug-susceptible PTB using cohort data of PTB cases newly registered in the Japan tuberculosis (TB) surveillance system between 2009 and 2018. Overall, 9151 overseas-born PTB cases were included in this study, and 70.3% were aged 34 years old or younger. “Students of high school and higher” (28.6%) and “regular workers other than service related sectors” (28.5%) accounted for over half of the study population, and they have continued to increase. Overall, the treatment success rate was 67.1%. Transferred-out constituted the largest proportion (14.8%) among the treatment non-success rate (32.9%). Multiple logistic regression analysis revealed patients whose health insurance type was “others and unknown”, including the uninsured (adjusted OR (AOR) = 3.43: 95% Confidence Intervals (CI) 2.57–4.58), those diagnosed as TB within “one year” (AOR = 2.61, 95% CI 1.97–3.46) and “1–5 years” (AOR = 2.44, 95% CI 1.88–3.17) of arrival in Japan, and males (AOR = 1.34, 95% CI 1.16–1.54), which were the main factors associated with treatment non-success. These findings imply that Japan needs to develop TB control activities considering the increasing trends of overseas-born PTB patients, the majority of whom are young and highly mobile. There is a need to pay greater attention to overseas-born PTB patients diagnosed within a short duration after entering Japan, who may be socially and economically disadvantaged for their treatment completion.

## 1. Introduction

Tuberculosis (TB) is one of the deadliest infectious diseases in the world. In 2020, an estimated 10 million people have developed TB and 1.5 million people died from TB worldwide [1]. To end TB, the End TB Strategy of the World Health Organization (WHO) urges rapid diagnosis, treatment, and care for everyone with TB, and especially for vulnerable populations such as migrants and refugees [2]. One of its goals is to achieve more than ninety percent of TB treatment success rate by 2025, however, international studies have generally reported poorer treatment success rates among migrants than native patients in various regions and countries [3,4,5].

Japan has become a low TB burden country, with a notification rate of 9.2 per 100,000 populations in 2021. While the notification rate has been decreasing steadily, both the number and proportion of overseas-born TB case have increased rapidly in the past decade, from 4.2% in 2010 to 11.3% in 2020 [6].

TB treatment outcomes among all age groups in Japan have been poorer among Japan-born than overseas-born patients, as the majority of Japan-born patients are elderly and the proportion of those who have died is high. However, a previous study has reported that TB treatment outcomes among those under 55 years old have been poorer among overseas-born than Japan-born patients [7]. Several studies have examined various aspects of treatment outcomes among overseas-born patients [8,9,10,11,12]. However, the demography of overseas-born people coming to and leaving Japan has undergone drastic changes in the past decade. This study therefore aimed to firstly describe the characteristics and treatment outcomes of, and secondly, identify the factors for treatment non-success among overseas-born pulmonary TB (PTB) patients in Japan over a period of ten years, using national surveillance data from the Japan TB Surveillance system (JTBS). According to the WHO guideline, treatment outcomes are evaluated separately for drug-susceptible and multi-drug resistant (MDR) patients, because both the treatment regimen and the definition of treatment outcomes are different [13]. As the notified number of MDR patients is still very limited in Japan [14], this study focused on drug-susceptible PTB patients.

## 2. Materials and Methods

### 2.1. Data Source—Japan TB Surveillance System

TB is a notifiable disease in Japan, and every physician who has diagnosed a case of TB is required to report to the local public health center (PHC) under Japan’s infectious disease control law. PHCs are responsible for entering and updating patient information to the electronic surveillance system, the JTBS. Under the JTBS, treatment outcomes are evaluated in the following year, and are released as a cohort dataset.

The JTBS regularly undergoes system revisions. In 2017, a major revision was made which changed the way treatment outcomes were evaluated. Treatment outcomes of those notified in 2015 were determined according to a computerized algorithm, which meant that only treatment outcomes of drug-susceptible pulmonary TB patients, who have received standardized treatment, were evaluated. However, with the revision, treatment outcomes of those notified in and after 2016 are entered manually by PHCs, allowing for evaluation of all types of TB.

### 2.2. Study Design and Populations

We conducted a nationwide retrospective cohort study on overseas-born persons diagnosed and notified with PTB to the JTBS between 1 January 2009 and 31 December 2018. All those persons who were notified as PTB and overseas-born were included. In order to allow comparison over a period of ten years, those who were diagnosed with MDR-TB, and exclusively with extrapulmonary TB, whose treatment outcomes were not available prior to 2016, were excluded from the study.

### 2.3. Definitions and Data Management

#### 2.3.1. Treatment Outcomes

For the purpose of this study, “cure” and “treatment completed” were combined as “treatment success”, and all the other treatment outcomes as “treatment non-success”, namely, “died”, “treatment failed”, “lost to follow-up (LTFU)”, “transferred-out”, “still on treatment”, and “unclassified” or “unknown” [15]. Transferred-out included both transfer within the country and transfer out of Japan. “Unclassified” included all those whose treatment could not be confirmed to be a standard treatment regimen for PTB, including those who received non-standard regimen, those who were notified after death, and those whose data were either insufficient or not entered. With the aforementioned revision, “unclassified” was replaced with “unknown”.

#### 2.3.2. Other Socio-Economic Variables

In JTBS, information regarding “nationality” for overseas-born patients has been collected since 2007. In 2012, the category of “nationality” was changed to “country of birth” [14]. For the purpose of this study, the two categories were treated as meaning the same. “Country of birth” was then further categorized into four income groups, as defined by the World Bank, i.e., low, lower-middle, upper-middle, and high-income countries. Income classifications as of 2018 were used, because the classification of the majority of the countries of birth of overseas-born patients hardly changed during 2009–2018 [16]. JTBS also collects the year of entry to Japan. Thus, “time between entry to Japan and TB diagnosis” was created based on the yearly calculation of year of entering Japan and the year diagnosed as TB in Japan. Occupational categories under JTBS include high school students and higher; workers in service industries, and health and educational sectors; regular workers other than service industries, and health and educational sectors; temporary and day workers; others, including unknown; unemployed. In this study, “service industry workers”, “physicians, nurses, and other healthcare workers”, and “teachers” were recategorized as one category of “workers in service industries, health and educational sectors”, and all the other regular i.e., full-time employees as “regular workers other than service industries, and health and educational sectors”. Unemployed include all those without regular and/or irregular jobs, including the elderly who have retired.

### 2.4. Statistical Analysis

Changes in the characteristics and treatment outcomes of overseas-born PTB patients were analyzed descriptively over a period of ten years. Due to the changes made to the definitions of treatment outcomes, as mentioned above, treatment outcomes were analyzed for ten years and for three different year periods (2009–2011, 2012–2015, and 2016–2018), respectively. In order to identify the factors for “treatment non-success” from health, socioeconomic, and migrant-related perspectives, the following variables were selected: (1) demographic variables: sex, age group; (2) clinical variables: HIV comorbidity, sputum smear results, culture result; (3) socioeconomic variables: occupation or social status category, health insurance type; (4) migrant-related variables: country of birth by income classifications, time between entry to Japan and TB diagnosis.

Univariate logistic regression analysis was used to estimate the association between each covariate and treatment non-success. Multiple logistic regression analysis was performed to estimate odds ratios (ORs) of treatment non-success. Adjusted OR (AOR) with a 95 percent confidence interval (95% CI) were calculated, and a *p*-value of less than 0.05 was considered statistically significant. All analyses were conducted using Stata version 16.1 (StataCorp, College Station, TX, USA).

## 3. Results

### 3.1. Characteristics of Overseas-Born Patients with PTB

Treatment results of a total of 9341 overseas-born cases with PTB who were notified between 2009 and 2018 were available from the JTBS cohort dataset. Overall, 190 cases of MDR were excluded, leaving a total of 9151 cases for analysis. Table 1 shows the characteristics of the patients for the entire study period and by treatment outcomes, and Table 2 shows the characteristics of the patients for the three periods. i.e., 2009–2011, 2012–2015, and 2016–2018. Among all patients (*n* = 9151), 70.3% (*n* = 6349) were aged 34 years old or younger, and the proportion of those under 24 years old has increased over the ten years. Overall, 48.1% (*n* = 4400) were born in lower middle-income countries, and 31.8% (*n* = 2906) were born in upper-middle income countries, however, the proportion of those born in lower middle-income countries increased drastically: from 37.4% in 2009–2011 to 57.8% in 2016–2018. In terms of the county of birth, the major countries of birth were China and the Philippines between 2009 and 2015. However, the number of Vietnamese-born patients surged in the mid-2010s and became the largest group in 2018. The number of patients from Indonesia, Nepal, and Myanmar started to increase as well (Figure 1). As for time between entry to Japan and diagnosis, 4479 of the 9151 patients for whom “the year of entering Japan” data were available were evaluated. Moreover, 25.2% (1129/4479) of cases were diagnosed as TB within one year of, and 48.3% (2164/4479) of cases within 1–5 years of entry to Japan. By occupation or social status category, two major groups accounted for over half of the study populations, namely, “students of high school and higher” (28.6%, 2620/9151) and “regular workers other than service industries, and health and educational sectors” (28.5%, 2607/9151), and they have continued to increase (Table 2). The number of both “students of high school and higher” and “regular workers other than service industries, and health and educational sectors” has more than doubled in ten years (Figure 2). In particular, the number of the latter increased drastically and became the largest group in 2018.

### 3.2. Treatment Outcomes

Figure 3 shows the treatment outcomes for the entire study period, and for the three periods. Over the entire study period, the treatment success rate was 67.1% (6143/9151), and the treatment non-success rate was 32.9% (3008/9151). Regarding the treatment non-success outcomes, transferred-out constituted the largest proportion (14.8%, 1353/9151), followed by unknown (6.3%, 576/9151), still on treatment (5.3%, 484/9151), LTFU (4.7%, 426/9151), died (1.6%, 147/9151), and treatment failed (0.2%, 22/9151). The proportion of treatment success has steadily increased from 61.9% (2009–2011) to 75.1% (2016–2018). On the contrary, the proportion of “unknown” and “LTFU” decreased noticeably: the former declined from 11.7% (2009–2011) and to 0.6% (2016–2018), and the latter from 6.6% (2009–2011) to 1.7% (2016–2018). The combined proportion of “died” and “treatment failed” was less than 2.0% throughout the total study period.

### 3.3. Factors of Treatment Non-Success

Table 3 summarizes the results of the multiple logistic regression analysis over the entire study period. Adjusted analysis indicated several risk factors for “treatment non- success”. Risk factors with the highest AOR were patients whose health insurance type were “others and unknown”, including those uninsured (AOR = 3.43: 95% CI 2.57–4.58), followed by patients with a medical care system for the elderly aged 75 years old and over (AOR = 2.90, 95% CI 1.38–6.07) and patients diagnosed as TB within one year (AOR = 2.61, 95% CI 1.97–3.46) of arrival in Japan.

Compared with the results of the univariate analysis, especially the AOR of both patients diagnosed as TB within one year and 1–5 years of arrival in Japan increased notably. Patients with national health insurance, which was not a significant risk factor in the univariate analysis, were observed as a risk factor by the adjusted analysis.

The risk factors changed somewhat over the three periods; however, the following were constantly identified as risk factors for treatment non-success; patients whose health insurance type was “others and unknown”, those diagnosed as TB within 5 years of arriving to Japan, and male (Table 4).

## 4. Discussion

This is the first study to describe the characteristics and treatment outcomes of, and identify the factors associated with, treatment non-success among overseas-born patients with drug-susceptible PTB at a national level in Japan over a relatively long period, i.e., ten years. Our study population was predominantly young adults, recent migrants, and those from middle income countries. Regarding treatment outcomes, “transferred-out” remained the main reason for treatment non-success over the study period. Socio-economic, migrant-related, and demographic factors increased the risk of treatment non-success outcomes.

The transition in the attributes of overseas-born patients with PTB over the decade to a certain extent reflects the changing pattern of immigration to Japan in the recent years. Thus, the increasing number of young patients from Southeast Asian countries such as Vietnam, Indonesia, Nepal, and Myanmar was to be expected as a result of a series of policy packages such as the “Technical Intern Training Program (TITP)” and the “300,000 International Student Policy”, which were launched in 1993 and 2008, respectively, to promote international cooperation and enhance international competitiveness [17,18]. The TITP was established to accept trainees from developing countries for a certain period and provide skills and knowledge in Japan. Labor-related laws have been applied to this program since 2010. The number of trainees has been constantly increasing, however, it has accelerated even further since the mid-2010s [17]. Most trainees are in their 20s [19], and usually stay in Japan for three to five years. Under the TITP, Vietnam, China, the Philippines, and Indonesia contributed to 90.9% of the sending countries in 2018 [20]. Many studies have pointed to socio-economic vulnerability and poorer health among such trainees in Japan [21,22,23,24], including TB patients who were forced to resign or return to their countries of origin due to TB [24,25,26]. Within the current JTBS, it is not possible to capture trainees under TITP. They are, however, mostly likely notified as “regular workers other than service industries, and health and educational sectors”, which has shown a remarkable increase after 2014.

The “300,000 International Student Policy”, which sought to invite 300,000 international students to Japanese universities and graduate schools by 2020, reached its numeric target in 2019. However, the majority of the increase was due to the increase in students of Japanese language schools, again from countries such as China, Vietnam, and Nepal [27]. It has been indicated that compared with international students of universities, students of Japanese language schools are less fluent in Japanese language, economically more disadvantaged, and less likely to receive a permanent job and continue to stay in Japan [28]. Again, it is impossible to specifically identify students of Japanese language schools from JTBS, however, numerous studies have reported TB outbreaks in Japanese language schools, and higher vulnerability towards TB infection and disease among such students [29,30,31].

The implications of these developments to TB control among overseas-born persons must be considered seriously. Unlike people migrating for settlement purpose, the majority of overseas-born people who are being diagnosed with TB in Japan are young, socio-economically disadvantaged, highly mobile, and hence likely to remain unfamiliar with the Japanese language, culture, and various social systems. Barriers to accessing healthcare are considerable, and a holistic approach, from early diagnosis to possibly continuity of care across international borders, is necessary throughout the entire migration process. At the same time, skills and capacity-strengthening for healthcare workers are also needed in Japan. Improvement of care and support is needed for migrant TB patients to complete their treatment completion.

Regarding treatment outcome, the apparent improvement in “treatment success” and the drastic decline in “LTFU” and “unknown” after 2016 is most likely due to the system revision of JTBS, and not a reflection of a real change. As previously mentioned, this is because since 2016, treatment outcomes have been entered manually by PHCs instead of a computed algorithm. However, what remained constant throughout the study period was the high proportion of “transferred out”, which has constantly been significantly higher than that among Japan-born patients [14]. As mentioned earlier, under JTBS, the “transfer out” includes both transfer within Japan and outside of Japan. A previous study specifically examining the “transfer out” cases in Japan indicated that compared with the Japan-born patients, the proportion of “international transfer-out” was considerably higher among overseas-born patients [11], implying a higher risk of treatment interruption [26], and potentially also of developing and spreading MDR-TB [32,33]. Globally, crossing international borders during TB treatment has been reported to pose a serious threat to the continuation and completion of TB treatment [34]. National and transnational programs to ensure continuity of care and enable evaluation of final treatment outcome of patients who cross international borders is therefore paramount. However, globally, such attempts are still limited [35]. In Japan, treatment outcomes of those who have transferred out of Japan remain unknown [36], however, recently, a research program has been started to coordinate cross-border referral for patients who were diagnosed as TB in Japan but who have decided to leave Japan to continue treatment elsewhere [37].

More caution is probably required to interpret domestic transfer out, which may or may not imply treatment interruption. Domestic transfer out includes both cases of patients moving within Japan and completing treatment in the new jurisdiction or becoming lost to follow-up during the course of the move. Further studies are needed to explore the potential risk factors for interruption of treatment among those who transfer out within Japan.

Four factors were identified as a risk for treatment non-success throughout the study periods; overseas-born patients whose health insurance category were “others and unknown” including the uninsured, those diagnosed as TB within one year and 1–5 years after arrival in Japan, and male.

Patients whose health insurance type was “other and unknown” were identified with the highest risk factor of treatment non-success. Among the patients under this category, the majority were originally categorized as “others” under the JTBS, most of whom are unlikely to be covered by any health insurance schemes and therefore paid their medical fees out of pocket. In Japan, all medical costs for TB treatment are subsidized during hospitalization under the infectious diseases control law, regardless of nationality. However, with the exception of TB-related drugs and examinations, outpatients have to bear a certain amount of medical costs to receive TB treatment and care, which could substantially increase if patients do not have any health insurance or receive social welfare assistance [38]. Previous studies have indicated that such a financial burden could discourage patients from accessing and/or continuing treatment, ultimately leading to poorer treatment outcomes [39,40].

Overseas-born patients who were diagnosed as TB within 5 years after arrival in Japan also had a higher risk of treatment non-success. Use of medical interpreter services by healthcare providers for overseas-born TB patients is still very limited [25], implying that for those overseas-born persons arriving to Japan with limited Japanese language skills, barriers to understanding and accessing not only health but general life and social welfare services can be enormous. Those who have recently arrived are also more likely to be socially disconnected compared to those who have been living in Japan for a longer period. These factors in combination could negatively influence their treatment outcomes [40]. Indeed, several studies from Japan have reported that overseas-born patients who were diagnosed relatively short after their arrival to Japan were more likely to be lost to follow-up [11], or transferred out [24].

Males had a higher risk of treatment non-success compared with females throughout the study year periods. Poorer treatment outcomes among males are commonly reported around the world [3,4,41], with a higher proportion of death, treatment failed, and LTFU among males compared to females. Various demographic, social, and cultural factors have been identified as possibly explaining the gender difference [41,42]. A further study in Japan may be necessary to explore the reasons why males in Japan are at a higher risk of poorer treatment outcomes.

Other variables, such as those unemployed, those with medical care system for elderly above 75 years, HIV comorbidity, and those born in lower-middle income, upper-middle income, and high-income countries, were also identified as risk factors of treatment non-success over the entire study period, but only in selected years in the sub-analysis year periods. This is probably due to the different characteristics of overseas-born people, depending on the study years of analysis. Therefore, these results must be interpreted with caution. According to international literature, old age has been reported as a higher risk factor of death among PTB patients [42,43]. HIV-TB co-infection is known to increase a risk of treatment failure [44]. In our study, patients born in lower-middle income, upper-middle income, and high-income countries had a higher risk of treatment non-success compared with patients born in low-income countries in the years 2016–2018 in the sub-analysis year period. Since being “transferred-out” was the major component of treatment non-success in our study, patients born in low-middle-income, upper-middle income, and high-income countries may have had more economic options to change their place of residence or leave Japan during treatment compared to patients born in low-income countries.

Diabetes mellitus (DM) has been known as a risk factor of treatment failure for PTB patients [43]. However, DM was excluded from our analysis due to the low rate of DM comorbidity among overseas-born PTB patients in Japan [45]. In addition, the definition of DM relies on the patient’s self-report under the JTBS. Homeless history variable was excluded in our study, because almost half of the study populations were categorized as “unknown on homeless history” in the original JTBS data. However, this fact implies that the assessment of overseas-born patients may be limited in many health facilities, probably because of language barriers [25]. Healthcare providers need to be encouraged to use medical interpreter services for the overseas-born TB patients to understand the health and social conditions of each migrant patient with limited Japanese proficiency. For that purpose, public medical interpreter services should be available nationwide in Japan [25].

As a strength of this study, we utilized nationwide data that were collected by PHCs in accordance with the standardized data entry procedures. Our results may therefore be considered representative of the characteristics and treatment outcomes of overseas-born TB patients in Japan.

This study is not without limitations. First, this study did not explore the risks of treatment non-success by different causes of treatment outcomes. We were therefore not able to identify which factors were associated with, for example, transferred out or LTFU. Second, due to the systematic limitation of utilizing the JTBS data, this study could not include variables which have previously been reported as potential risk factors of unfavorable treatment outcomes, such as smoking [46], alcohol addiction [47], education level [44,46], and lack of family and social support [42]. Third, MDR-TB patients, who have been reported as high risk factors of unsuccessful treatment outcomes [3], were not included in this study, because treatment outcomes need to be evaluated separately for drug-susceptible and MDR patients. In addition, the number of MDR-TB patients was very limited.

## 5. Conclusions

The characteristics of the overseas-born PTB patients in Japan have remarkably changed over the past decade, with increasing numbers of patients who are young, socio-economically disadvantaged, and highly mobile. Due to the improvement of the JTBS system, treatment success rate increased, but transferred-out remained the major reason for treatment non-success outcomes throughout the study period.

Various socio-economic, migrant-related, and demographic factors could possibly increase the risk of poorer treatment outcomes among overseas-born PTB patients. Therefore, a holistic approach is required to ensure that such patients are able to access the care they need, both while they are in Japan and elsewhere, if and when they decide to leave Japan while they are still on treatment for TB. Today, more than ever, Japan relies on immigration to solve a number of its social and economic issues. It is therefore paramount that Japan invests in TB control not only within Japan, but also in the sending countries via international cooperation, to end TB.

## Figures and Tables

**Figure 1 ijerph-19-12598-f001:**
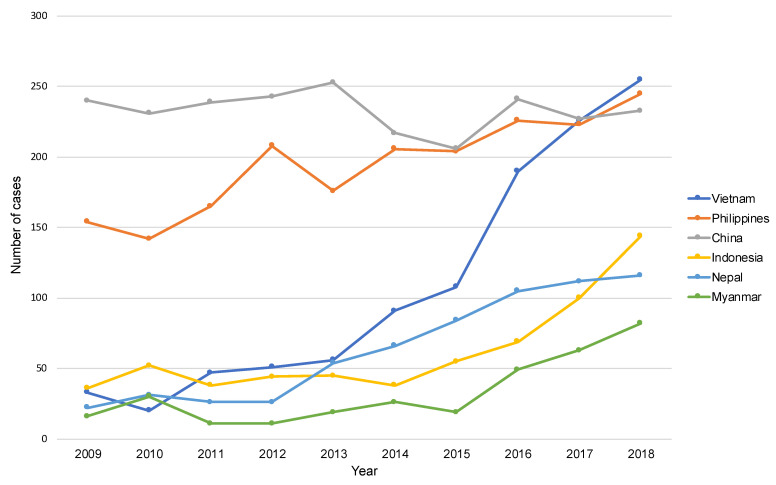
The number of overseas-born patients with pulmonary tuberculosis by selected countries of birth in 2009–2018.

**Figure 2 ijerph-19-12598-f002:**
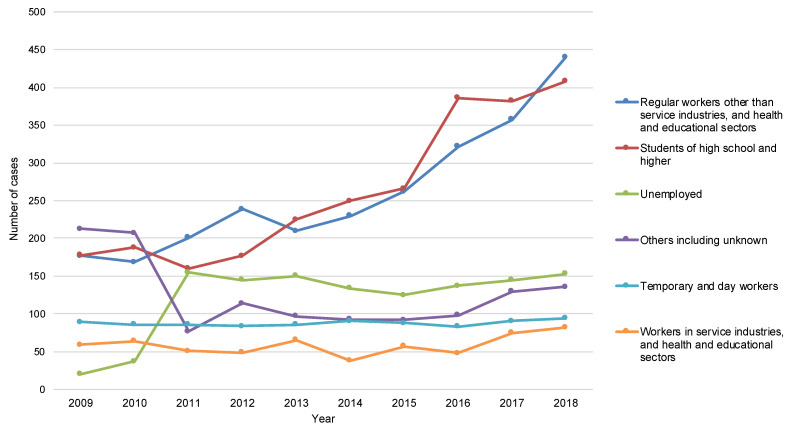
The number of overseas-born patients with pulmonary tuberculosis by occupation or social status in 2009–2018.

**Figure 3 ijerph-19-12598-f003:**
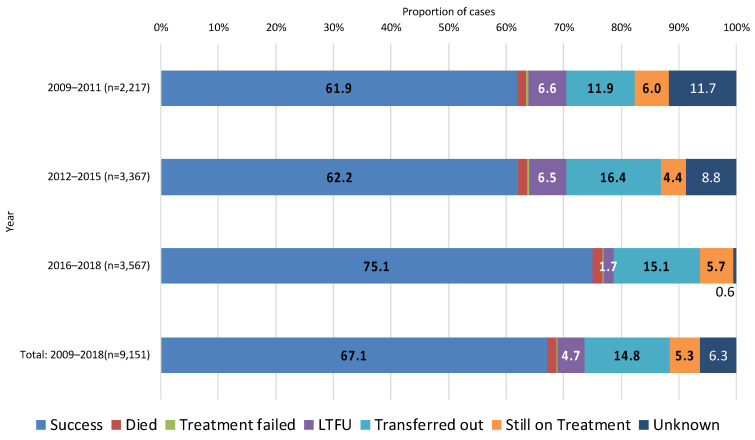
Treatment outcome of overseas-born patients with pulmonary tuberculosis in 2009–2018, and by the three periods. Abbreviations: LTFU, Lost to follow-up. Notes: Success includes “cure”, and “completed”.

**Table 1 ijerph-19-12598-t001:** Characteristics of overseas-born patients with pulmonary tuberculosis in 2009–2018, by treatment outcomes.

Category	Sub-Category	Treatment Success *n* = 6143 (67.1%)		Non-Success *n* = 3008 (32.9%)		Total *n* = 9151	
		*n*	%	*n*	%	*n*	%
Sex	Female	3203	70.0	1371	30.0	4574	50.0
	Male	2940	64.2	1637	35.8	4577	50.0
Age group	Under 24	2130	67.8	1011	32.2	3141	34.3
	25–34	2281	69.2	1017	30.8	3298	36.0
	35–44	865	69.1	387	30.9	1252	13.7
	45–54	476	68.2	222	31.8	698	7.6
	55–64	196	59.2	135	40.8	331	3.6
	65+	195	45.2	236	54.8	431	4.7
Country of birth by income classification	Low income country	531	74.3	184	25.7	715	7.8
	Lower-middle income country	3024	68.7	1376	31.3	4400	48.1
	Upper-middle income country	1892	65.1	1014	34.9	2906	31.8
	High-income country	411	61.3	260	38.8	671	7.3
	Unknown country of birth	285	62.1	174	37.9	459	5.0
Time between entry to Japan and TB diagnosis	10+ year	536	74.8	181	25.2	717	16.0
	Less than 1 year	751	66.5	378	33.5	1129	25.2
	1–5 year	1523	70.4	641	29.6	2164	48.3
	5–10 year	368	78.5	101	21.5	469	10.5
Occupation or social status	Students of high school and higher	1852	70.7	768	29.3	2620	28.6
	Workers in service industries, and health and educational sectors *	415	70.6	173	29.4	588	6.4
	Regular workers other than service industries, and health and educational sectors	1825	70.0	782	30.0	2607	28.5
	Temporary and day workers	590	67.2	288	32.8	878	9.6
	Others, including unknown **	784	62.4	473	37.6	1257	13.7
	Unemployed	677	56.4	524	43.6	1201	13.1
Health insurance type	Employees’ health insurance	2293	72.0	890	28.0	3183	34.8
	National health insurance	3271	70.3	1382	29.7	4653	50.8
	Medical care system for the elderly aged 75 and over	66	38.4	106	61.6	172	1.9
	Social welfare assistance ***	163	57.8	119	42.2	282	3.1
	Others and unknown	350	40.7	511	59.4	861	9.4
HIV comorbidity	Not-positive ****	4749	69.0	2136	31.0	6885	99.3
	Positive	21	42.9	28	57.1	49	0.7
Sputum smear	Negative, not tested, and unknown	3910	68.6	1794	31.5	5704	62.3
	Positive	2233	64.8	1214	35.2	3447	37.7
Culture	Negative	2098	71.6	833	28.4	2931	32.0
	Positive	3636	68.0	1715	32.1	5351	58.5
	Others	409	47.1	460	52.9	869	9.5
2009–2011		1373	61.9	844	38.1	2217	24.2
2012–2015		2093	62.2	1274	37.8	3367	36.8
2016–2018		2677	75.1	890	25.0	3567	39.0
Total (2009–2018)		6143	67.1	3008	32.9	9151	100.0

Abbreviations: HIV, human immunodeficiency virus; TB, tuberculosis. Notes: Treatment success includes “cure”, and “completed”. Non-success includes “died”, “treatment failed”, “lost to follow-up”, “transferred-out”, “still on treatment”, and “unknown”. “Time between entry to Japan and TB diagnosis” and “HIV” data are available after year 2012. Each subject number is 4479, and 6934 respectively. * “Workers in service industries, and health and educational sectors” include service industry workers, health personnel, long term care workers, teachers, and childminders. ** “Others including unknown” includes all infants, children, and students of junior high schools and younger, self-employed, house workers, others and unknown. *** “Social welfare assistance” includes a person who currently applies for social welfare assistance. **** “Not-positive” includes negative, not tested, and unknown.

**Table 2 ijerph-19-12598-t002:** Characteristics of overseas-born patients with pulmonary tuberculosis by year periods.

Category	Sub-Category	2009–2011 (*n* = 2217)		2012–2015 (*n* = 3367)		2016–2018 (*n* = 3567)	
		*n*	%	*n*	%	*n*	%
Sex	Female	1202	54.2	1701	50.5	1671	46.9
	Male	1015	45.8	1666	49.5	1896	53.2
Age group	Under 24	681	30.7	1102	32.7	1358	38.1
	25–34	833	37.6	1171	34.8	1294	36.3
	35–44	334	15.1	502	14.9	416	11.7
	45–54	168	7.6	289	8.6	241	6.8
	55–64	75	3.4	127	3.8	129	3.6
	65+	126	5.7	176	5.2	129	3.6
Country of birth by income classification	Low income country	114	5.1	248	7.4	353	9.9
	Lower-middle income country	828	37.4	1512	44.9	2060	57.8
	Upper-middle income country	902	40.7	1145	34.0	859	24.1
	High-income country	263	11.9	257	7.6	151	4.2
	Unknown country of birth	110	5.0	205	6.1	144	4.0
Time between entry to Japan and TB diagnosis	10+ year	-	-	393	18.7	324	13.6
	Less than 1 year	-	-	497	23.7	632	26.6
	1–5 year	-	-	949	45.2	1215	51.1
	5–10 year	-	-	260	12.4	209	8.8
Occupation or social status	Students of high school and higher	526	23.7	918	27.3	1176	33.0
	Workers in service industries, and health and educational sectors *	174	7.9	209	6.2	205	5.8
	Regular workers other than service industries, and health and educational sectors	547	24.7	941	28.0	1119	31.4
	Temporary and day workers	261	11.8	349	10.4	268	7.5
	Others, including unknown **	497	22.4	396	11.8	364	10.2
	Unemployed	212	9.6	554	16.5	435	12.2
Health insurance type	Employees’ health insurance	680	30.7	1169	34.7	1334	37.4
	National health insurance	1096	49.4	1703	50.6	1854	52.0
	Medical care system for the elderly aged 75 and over	56	2.5	72	2.1	44	1.2
	Social welfare assistance ***	93	4.2	116	3.5	73	2.1
	Others and unknown	292	13.2	307	9.1	262	7.4
HIV comorbidity	Not positive ****	-	-	3340	99.2	3545	99.4
	Positive	-	-	27	0.8	22	0.6
Sputum smear	Negative, not tested, and unknown	1328	59.9	2066	61.4	2310	64.8
	Positive	889	40.1	1301	38.6	1257	35.2
Culture	Negative	670	30.2	1077	32.0	1184	33.2
	Positive	1230	55.5	1945	57.8	2176	61.0
	Others	317	14.3	345	10.3	207	5.8

Abbreviations: HIV, human immunodeficiency virus; TB, tuberculosis. Notes: “Time between entry to Japan and TB diagnosis” and “HIV” data are available after year 2012. Each subject number is 4479, and 6934, respectively. * “Workers in service industries, and health and educational sectors” include service industry workers, health personnel, long term care workers, teachers, and childminders. ** “Others including unknown” includes all infants, children, and students of junior high schools and younger, self-employed, house workers, others, and unknown. *** “Social welfare assistance” includes a person who currently applies for social welfare assistance. **** “Not-positive” includes negative, not tested, and unknown.

**Table 3 ijerph-19-12598-t003:** Multiple logistic regression of risk factors associated with treatment non-success among overseas-born patients with pulmonary tuberculosis in 2009–2018.

Category	Sub-Category	Univariate (Unadjusted)				Multivariable AOR			
		OR	95% CI		*p*-Value	OR	95% CI		*p*-Value
Sex	Female	1.0				1.0			
	Male	1.30	1.19	1.42	<0.001	1.34	1.16	1.54	<0.001
Age group	Under 24	1.0				1.0			
	25–34	0.94	0.85	1.04	0.24	0.98	0.83	1.16	0.82
	35–44	0.94	0.82	1.09	0.41	1.17	0.91	1.51	0.23
	45–54	0.98	0.82	1.17	0.85	1.35	0.96	1.90	0.09
	55–64	1.45	1.15	1.83	0.002	1.53	0.98	2.38	0.06
	65+	2.55	2.08	3.13	<0.001	1.59	0.94	2.68	0.08
Country of birth by income classification	Low income country	1.0				1.0			
	Lower-middle income country	1.31	1.10	1.57	0.003	1.50	1.16	1.95	0.002
	Upper-middle income country	1.55	1.29	1.86	<0.001	1.63	1.25	2.14	<0.001
	High-income country	1.83	1.45	2.29	<0.001	1.53	1.01	2.32	0.04
	Unknown country of birth	1.76	1.37	2.27	<0.001	1.47	0.87	2.49	0.15
Time between entry to Japan and TB diagnosis	10+ year	1.0				1.0			
	Less than 1 year	1.49	1.21	1.84	<0.001	2.61	1.97	3.46	<0.001
	1–5 year	1.25	1.03	1.51	0.03	2.44	1.88	3.17	<0.001
	5–10 year	0.81	0.62	1.07	0.14	1.24	0.91	1.70	0.17
Occupation or social status	Students of high school and higher	1.0				1.0			
	Workers in service industries, and health and educational sectors *	1.01	0.83	1.22	0.96	1.49	1.05	2.11	0.03
	Regular workers other than service industries, and health and educational sectors	1.03	0.92	1.16	0.59	1.37	1.09	1.74	0.01
	Temporary and day workers	1.18	1.00	1.39	0.051	1.53	1.15	2.04	0.003
	Others, including unknown **	1.45	1.26	1.68	<0.001	1.66	1.26	2.19	<0.001
	Unemployed	1.87	1.62	2.15	<0.001	2.06	1.55	2.74	<0.001
Health insurance type	Employees’ health insurance	1.0				1.0			
	National health insurance	1.09	0.99	1.20	0.10	1.32	1.09	1.60	0.004
	Medical care system for the elderly aged 75 and over	4.14	3.01	5.68	<0.001	2.90	1.38	6.07	0.01
	Social welfare assistance ***	1.88	1.47	2.41	<0.001	1.56	0.92	2.66	0.100
	Others and unknown	3.76	3.22	4.40	<0.001	3.43	2.57	4.58	<0.001
HIV comorbidity	Not positive ****	1.0				1.0			
	Positive	2.96	1.68	5.23	<0.001	2.21	1.00	4.87	0.05
Sputum smear	Negative, not tested, and unknown	1.0				1.0			
	Positive	1.18	1.08	1.30	<0.001	1.19	1.02	1.39	0.03
Culture	Negative	1.0				1.0			
	Positive	1.19	1.08	1.31	0.001	1.14	0.97	1.36	0.12
	Others	2.83	2.42	3.31	<0.001	2.38	1.83	3.10	<0.001

Multivariable models adjusted for sex, age, sputum smear, and culture. Abbreviations: AOR, adjusted odds ratio; CI, confidence interval; HIV, human immunodeficiency virus; OR, odds ratio; TB, tuberculosis. Notes: The number and the percentage of treatment outcomes are referred to in Table 1. “Time between entry to Japan and TB diagnosis” and “HIV” data are available after year 2012. Each subject number is 4479, and 6934 respectively. * “Workers in service industries, and health and educational sectors” include service industry workers, health personnel, long term care workers, teachers, and childminders. ** “Others including unknown” includes all infants, children, and students of junior high schools and younger, self-employed, house workers, others, and unknown. *** “Social welfare assistance” includes a person who currently applies for social welfare assistance. **** “Not-positive” includes negative, not tested, and unknown.

**Table 4 ijerph-19-12598-t004:** Multiple logistic regression of risk factors associated with treatment non-success among overseas-born patients with pulmonary tuberculosis by year periods.

Category	Sub-Category	Year 2009–2011				Year 2012–2015				Year 2016–2018			
		AOR	95% CI		*p*-Value	AOR	95% CI		*p*-Value	AOR	95% CI		*p*-Value
Sex	Female	1.0				1.0				1.0			
	Male	1.24	1.02	1.50	0.03	1.25	1.03	1.52	0.03	1.51	1.22	1.85	<0.001
Age group	Under 24	1.0				1.0				1.0			
	25–34	0.83	0.66	1.05	0.12	0.82	0.65	1.04	0.10	1.22	0.96	1.54	0.11
	35–44	0.87	0.64	1.20	0.40	1.04	0.73	1.47	0.83	1.34	0.91	1.99	0.14
	45–54	0.62	0.42	0.94	0.02	1.57	0.99	2.51	0.06	1.06	0.61	1.85	0.84
	55–64	1.32	0.77	2.25	0.31	1.76	0.95	3.25	0.07	1.67	0.84	3.32	0.15
	65+	1.18	0.66	2.12	0.58	2.55	1.23	5.30	0.01	1.04	0.46	2.34	0.93
Country of birth by income classification	Low income country	1.0				1.0				1.0			
	Lower-middle income country	0.96	0.63	1.47	0.85	1.17	0.80	1.70	0.41	2.00	1.37	2.93	<0.001
	Upper-middle income country	0.93	0.61	1.41	0.72	1.30	0.89	1.89	0.17	1.73	1.14	2.61	0.01
	High-income country	0.88	0.55	1.43	0.61	1.08	0.62	1.88	0.80	2.01	1.05	3.82	0.03
	Unknown country of birth	1.32	0.76	2.31	0.33	1.45	0.75	2.80	0.27	1.01	0.38	2.69	0.99
Time between entry to Japan and TB diagnosis	10+ year	—	—	—	—	1.0				1.0			
	Less than 1 year	—	—	—	—	2.88	1.96	4.22	<0.001	3.17	2.01	4.99	<0.001
	1–5 year	—	—	—	—	2.38	1.68	3.37	<0.001	3.26	2.11	5.05	<0.001
	5–10 year	—	—	—	—	1.31	0.88	1.94	0.178	1.25	0.72	2.18	0.43
Occupation or social status	Students of high school and higher	1.0				1.0				1.0			
	Workers in service industries, and health and educational sectors *	0.84	0.55	1.28	0.41	1.91	1.18	3.09	0.01	1.19	0.69	2.05	0.53
	Regular workers other than service industries, and health and educational sectors	1.05	0.76	1.44	0.78	1.52	1.09	2.10	0.01	1.32	0.93	1.87	0.12
	Temporary and day workers	1.00	0.70	1.44	0.98	1.83	1.25	2.66	0.002	0.98	0.61	1.58	0.93
	Others, including unknown **	1.09	0.79	1.53	0.59	1.94	1.32	2.86	0.001	1.48	0.98	2.25	0.07
	Unemployed	1.35	0.90	2.02	0.14	2.10	1.42	3.12	<0.001	2.15	1.40	3.31	<0.001
Health insurance type	Employees’ health insurance	1.0				1.0				1.0			
	National health insurance	1.0	0.75	1.22	0.73	1.1	0.84	1.39	0.55	1.7	1.3	2.3	0.001
	Medical care system for the elderly aged 75 and over	2.0	0.91	4.32	0.09	1.9	0.65	5.63	0.24	5.9	2.0	17.5	0.001
	Social welfare assistance ***	1.5	0.89	2.40	0.13	1.1	0.54	2.22	0.81	2.8	1.2	6.4	0.02
	Others and unknown	2.1	1.58	2.88	<0.001	2.5	1.70	3.74	<0.001	4.7	3.0	7.4	<0.001
HIV comorbidity	Not-positive ****	—	—	—	—	1.0				1.0			
	Positive	—	—	—	—	3.67	1.17	11.52	0.03	1.12	0.32	3.93	0.86
Sputum smear	Negative, not tested and unknown	1.0				1.0				1.0			
	Positive	0.91	0.74	1.12	0.37	1.08	0.87	1.34	0.51	1.26	1.00	1.60	0.054
Culture	Negative	1.0				1.0				1.0			
	Positive	1.11	0.88	1.39	0.40	1.18	0.94	1.50	0.16	1.18	0.92	1.52	0.20
	Others	2.61	1.96	3.48	<0.001	2.32	1.65	3.25	<0.001	2.15	1.39	3.33	0.001

Multivariable models adjusted for sex, age, sputum smear, and culture. Abbreviations: AOR, adjusted odds ratio; CI, confidence interval; HIV, human immunodeficiency virus; OR, odds ratio; TB, tuberculosis. Notes: “Time between entry to Japan and TB diagnosis” and “HIV” data are available after year 2012. * “Workers in service industries, and health and educational sectors” include service industry workers, health personnel, long term care workers, teachers, and childminders. ** “Others including unknown” includes all infants, children, and students of junior high schools and younger, self-employed, house workers, others, and unknown. *** “Social welfare assistance” includes a person who currently applies for social welfare assistance. **** “Not-positive” includes negative, not tested, and unknown.

## Data Availability

The surveillance data supported the results of this study are available from the Tuberculosis Surveillance Center (https://jata-ekigaku.jp/english, accessed on 20 August 2022), but limits apply to the availability of raw data due to license issues. However, aggregated data used in this study might be available upon reasonable request with permission of the Tuberculosis Surveillance Center through contact with the author.

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
