# Peer review of "Characteristics and Treatment Outcomes among Migrants with Pulmonary Tuberculosis: A Retrospective Cohort Study in Japan, 2009–2018"

_ijerph, 2022, doi:10.3390/ijerph191912598_

Round 1
Reviewer 1 Report
Interesting study conveying the main characteristics of migrants in Japan including socioeconomic criteria. Omics are not yet available I guess.
Author Response
Thank you, please see the attachment.

Reviewer 2 Report
After careful reading and analysis, I consider the work to be well written, data satisfactorily collected and analyzed, and discussions and conclusions consistent with those proposed by the authors. I just recommend putting a dash in the case of "empty spaces" (no data) in the tables and re-evaluating table 4, it looks like there is some formatting issue, the data looks out of place.
Reviewer 3 Report
In this manuscript ‘Characteristics and Treatment Outcomes among Migrants With Pulmonary Tuberculosis: A Retrospective Cohort Study in Japan, 2009-2018,’ Lee et al conduct a retrospective analysis among 9,151 overseas-born patients with drug-susceptible pulmonary TB using cohort data of Japan TB surveillance system between 2009 and 2018, describing characteristics and treatment outcomes of this cohort. Treatment success, defined as ‘cured’ plus ‘treatment completed,’ was 67.1%. Variables associated with unfavorable treatment outcomes (per multivariate regression analysis) included individuals who were male, uninsured, and diagnosed as TB within 5 years of arrival in Japan.
Overall, the manuscript is well-summarized. One of the biggest strengths of this study is the large sample size. While this paper has certainly been engaging, there are some issues in this paper that require explanation. Please see comments below for specific concerns.
· The term non-treatment success is not used commonly and can be misinterpreted. Authors may consider replacing ‘non-treatment success’ with ‘not achieving treatment success’ OR ‘having unfavorable treatment outcomes’ everywhere in the manuscript.
· The occupation categories are very broadly classified. Is there data on specific occupation of patients? Health sector is different from educations sector which is again different from other services. Also, all ‘infants, children and students of junior high schools’ (pediatric population) is completely different from ‘self-employed or house workers’ and should not be grouped together under ‘others including unknown’. If these categories cannot be made narrower, figure 2 would be less meaningful and authors may consider removing it.
Minor comments:
Abstract
Line 22: please restructure the first sentence to make it clearer.
1. Introduction
Line 55: would remove ‘when stratified by age groups’ as the sentence is about migrants vs native patients.
2. Methods
Line 93: please explain how ‘still on treatment’ can be classified as ‘non-treatment success’. Can the authors exclude these patients as they’re neither considered treatment success nor those who did not achieve treatment success?
Line 126: Would consider rephrasing ‘biological’ with ‘demographic’ here and at other places.
3. Results
Line 141: For table 1, following ‘entire study period’, would also add the phrase ‘categorized by favorable and unfavorable treatment outcomes’
4. Discussion
Line 287: ‘system revision of JTBS?’ -please elaborate
It appears that migrants from lower-middle or upper-middle income countries appear to have more unfavorable outcomes compared to migrants from low-income countries – this different was significant even on adjusted analysis, hence merits explanation in the discussion.
Line 366: This study is limited to only drug-susceptible TB patients. Under limitations, would also add that MDR-TB patients who tend to do worse, were not included in this cohort.
5. Figures/Tables
Tables 1 - 4
· HIV comorbidity – how is ‘except for positive’ different from ‘negative’, if not different, would just say ‘negative’
· Sputum smear – would mention ‘not tested and unknown’ a category separate from ‘negative’
Figure 3: ‘2016-2018’ near ‘still on treatment’ and ‘unknown’ there is a stroke of line entered by error that can be removed
Author Response
Thank you, please see the attachment.
